# Detecting Forest Musk Deer Abscess Disease Pathogens Using 16S rRNA High-Throughput Sequencing Technology

**DOI:** 10.3390/ani13193142

**Published:** 2023-10-08

**Authors:** Guanjie Lu, Zhe Wang, Baofeng Zhang, Zhichao Zhou, Defu Hu, Dong Zhang

**Affiliations:** 1School of Ecology and Nature Conservation, Beijing Forestry University, Beijing 100083, China; luguanjie2023@163.com (G.L.); wzhe0314@163.com (Z.W.); baofengdaisy@126.com (B.Z.); zhouzhichao2018@163.com (Z.Z.); hudf@bjfu.edu.cn (D.H.); 2State Key Laboratory of Efficient Production of Forest Resources, Beijing Forestry University, Beijing 100083, China

**Keywords:** forest musk deer, abscess, pathogenic bacteria, 16S rRNA high-throughput sequencing technology

## Abstract

**Simple Summary:**

Abscess disease is a major disease that affects forest musk deer populations. Accurately identifying the types of pathogenic bacteria responsible for it is crucial for effective clinical treatment and the development of drugs and vaccines. This study is the first to use high-throughput 16S rRNA sequencing technology to detect the types and abundance of pathogenic bacteria in abscess disease samples at the genetic level, thereby overcoming limitations of previous methods. Microbial structure and bacterial community correlation analyses of six sequencing samples revealed that the dominant pathogenic bacteria were relatively singular and had an overwhelming abundance in the same individual. The pathogenic bacterial species differed among different individuals, and the dominant pathogenic bacteria exhibited no significant correlation with other bacteria in the pus, thus indicating that the dominant pathogenic bacteria were responsible for the production of abscess disease. The primary dominant pathogenic bacteria were *Trueperella pyogenes*, *Fusobacterium necrophorum*, and *Bacteroides fragilis*. While *Trueperella pyogenes* has been confirmed as one of the pathogenic bacteria responsible for abscess disease in forest musk deer, *Fusobacterium necrophorum* and *Bacteroides fragilis* could not be isolated or identified by previous research methods due to their obligate anaerobic characteristics. Therefore, this study is the first to report that *Fusobacterium necrophorum* and *Bacteroides fragilis* are the dominant pathogenic bacteria responsible for abscess disease in forest musk deer.

**Abstract:**

Currently, researchers use bacterial culture and targeted PCR methods to classify, culture, and identify the pathogens causing abscess diseases. However, this method is limited by factors such as the type of culture medium and culture conditions, making it challenging to screen and proliferate many bacteria effectively. Fortunately, with the development of high-throughput sequencing technology, pathogen identification at the genetic level has become possible. Not only can this approach overcome the limitations of bacterial culture, but it can also accurately identify the types and relative abundance of pathogens. In this study, we used high-throughput sequencing of 16S rRNA to identify the pathogens in purulent fluid samples. Our results not only confirmed the presence of the main pathogen reported by previous researchers, *Trueperella pyogenes*, but also other obligate anaerobes, *Fusobacterium necrophorum* and *Bacteroides fragilis* as the dominant pathogens causing abscess diseases for the first time. Therefore, our findings suggest that high-throughput sequencing technology has the potential to replace traditional bacterial culture and targeted PCR methods.

## 1. Introduction

The forest musk deer (*Moschus berezovskii*) is a solitary ruminant mammal, and the musk secreted by the scent glands of male individuals is a valuable traditional Chinese medicine and an important raw material for producing high-end perfumes [1]. In modern times, due to habitat changes and human interference, the forest musk deer population is on the brink of extinction. It has been listed in the Appendix of the Convention on International Trade in Endangered Species of Wild Fauna and Flora (CITES) and the International Union for Conservation of Nature (IUCN) Red List of Threatened Species. It is also a first-class protected species under the China’s Wildlife Protection Law. The Chinese government decided in the 1950s to encourage forest musk deer farming in order to protect and expand the population of forest musk deer [2]. Since the beginning of forest musk deer farming in 1953, diseases have been one of the main obstacles to the development of captive breeding. Abscess disease is one of the major diseases affecting captive forest musk deer [3].

The abscess disease of forest musk deer can occur all year round and affects forest musk deer of all age groups. Skin abscesses are mostly observed around the hooves, eye sockets, and other facial areas, while the pericardium, lungs, liver, and other internal organs are also frequently affected by abscess disease [4]. The surgical excision of skin abscesses combined with antibiotic treatment typically yields a favorable prognosis, but infections and abscesses within internal organs are difficult to detect and typically cannot be treated, ultimately leading to death. Currently, the primary clinical treatment method for abscesses involves using broad-spectrum antibiotics. Accurate identification of the pathogenic bacteria is essential to ensure proper use of antibiotic drugs in clinical treatment. Although researchers have isolated and identified some of the pathogenic bacteria causing abscesses, there are discrepancies among different studies regarding the types of bacteria reported [5,6,7,8]. The predominant factor contributing to this issue is that diagnosis has previously required isolation, cultivation, and identification. Nonetheless, pathogen cultivation is constrained by media type and cultivation conditions, which renders it challenging to screen and identify numerous bacteria effectively [9]. Thus, discovering a methodology that can surmount the constraints of bacterial cultivation and facilitate prompt and precise pathogenic bacterial identification is an essential prerequisite for accurate diagnosis and effective treatment of abscesses in the forest musk deer gland.

In recent years, with the development of high-throughput sequencing technology, the microbiome of clinical samples has been studied in depth, and its taxonomic composition has been determined [9]. In the diagnosis of pathogens, analyzing pathogens at the genetic level can overcome the limitations and time constraints associated with bacterial culture. Furthermore, this method is not impacted by antibiotic treatment, exhibits high sensitivity, and can fully identify the type and abundance of pathogens present [10]. At present, metagenomic sequencing is still very expensive, while 16S rRNA sequencing may be a relatively fast and economical alternative method [11]. Studies have used the 16S rRNA sequencing method to detect pathogenic bacteria causing mastitis in cows. In addition to identifying the recognized pathogens associated with mastitis, some additional types of pathogenic bacteria were also discovered [12].

In this study, we used 16S rRNA sequencing technology to analyze the bacterial content from several pus samples obtained from different sites on farmed forest musk deer abscesses. We analyzed the bacterial community composition and correlation structure in the pus samples, preliminarily identified the types of pathogens and their interrelationships, and further evaluated their potential use as a clinical diagnostic tool.

## 2. Materials and Methods

### 2.1. Pus Sample Collection

Between July 2021 and January 2022, 6 pus samples were collected from 5 adult forest musk deer (2–4 years old) with abscess in Fengxian County, Baoji City, Shaanxi Province, China (33°54′30″ N, 106°30′57″ E). Four of them were taken from different individuals while the other two were taken from different parts of the same forest musk deer. The ear tag number was used to identify the individual. Three samples were taken from individuals with abscesses on their body surface (Figure 1A), and three samples were taken from individuals that had died due to abscess disease and were subsequently dissected. Sample information is presented in Table 1. The superficial pus was collected by surgical incision of the pustule with the help of a resident veterinarian who cut the pustules after aseptic treatment and sucked the pus with a disposable medical syringe immediately after the pus flowed out for 1–2 s, after which it was transferred to sterile cryovials and stored in liquid nitrogen. Visceral pus was obtained from recently deceased forest musk deer through dissection following aseptic procedures. Pus with obvious purulence or diffuse pus was found in the lungs (Figure 1B). Surface pus from lung tissue with abscesses or tissue with abundant pus was aspirated using a disposable medical syringe and transferred to sterile cryovials for liquid nitrogen preservation. All samples were stored in dry ice and transferred to the laboratory and stored in a −80 °C freezer until DNA extraction.

All samples were collected during surgical diagnosis and treatment by the resident veterinarian and the ethical approval is not required.

### 2.2. DNA Extraction and Amplification

Microbial community total DNA was extracted using the E.Z.N.A.^®^ soil DNA kit (Omega Bio-tek, Norcross, GA, USA) following the manufacturer’s instructions. The quality of DNA extraction was checked using 1% agarose gel electrophoresis, and DNA concentration and purity were determined using NanoDrop2000. PCR amplification of the V3-V4 variable region of the 16S rRNA gene was performed using 338F (5′-ACTCCTACGGGAGGCAGCAG-3′) and 806R (5′-GGACTACHVGGGTW TCTAAT-3′). The amplification program was as follows: predenaturation at 95 °C for 3 min, 27 cycles (denaturation at 95 °C for 30 s, annealing at 55 °C for 30 s, amplification at 72 °C for 30 s), then stable extension at 72 °C for 10 min, and finally storage at 4 °C (PCR instrument: ABI GeneAmp ^®^ Type 9700, Applied Biosystems, Waltham, MA, USA). The PCR reaction system was: 5 × TransStart FastPfu buffer 4 μL, 2.5 mM dNTPs 2 μL, forward primer (5 μM) 0.8 μL, reverse primer (5 μM) 0.8 μL, *TransStart FastPfu* DNA polymerase 0.4 μL, and template DNA 10 ng 12 μL, making up to 20 μL.

### 2.3. 16S rRNA Gene Amplicon Sequencing and Data Processing

First, PCR amplification products were purified, quantified and standardized to form sequencing libraries. The constructed libraries were then checked for quality, and qualified libraries were sequenced on an Illumina NovaSeq 6000. The original image data file obtained by the final sequencing was converted into the original sequence through base-calling analysis and stored in FASTQ file format. The resulting files were uploaded to the NCBI database (Accession Number: PRJNA944830). Raw reads obtained from sequencing are filtered using Trimmomatic v0.33 software [13]. Cutadapt 1.9.1 software was then used to identify and remove primer sequences, resulting in clean reads that did not contain primer sequences [14]. The dada2 method in QIIME2 2020.6 was used to denoise [15,16], assemble double-ended sequences, and remove chimeric sequences to obtain the final valid data.

### 2.4. Statistical Analysis

The qualified sequences with more than 97% similarity thresholds were allocated to one operational taxonomic unit (OTU). Using SILVA as a reference database [17], the Naive Bayesian classifier is used to taxonomically annotate the feature sequence, and the species classification information corresponding to each feature can be obtained, and then the community composition of each sample is counted at each taxonomic level. QIIME software (2023.05) was used to generate species abundance tables at different taxonomic levels [15], and R language tools were used to draw community structure maps of samples at different taxonomic levels. According to the abundance and variation of each species in each sample, Spearman’s rank correlation analysis was performed and the data with a correlation greater than 0.1 and a *p* value less than 0.05 were screened to construct a correlation network.

## 3. Results

### 3.1. Sequence Statistics

Three of the nine samples collected and sequenced may have been contaminated during collection and were therefore excluded from the dataset before further analysis. The other six samples were sequenced by Illumina NovaSeq 6000, and a total of 480,852 pairs of original reads were obtained. A total of 479,654 clean reads were generated after quality control and splicing of paired-end reads, and at least 79,694 clean reads were generated for each sample. Clean reads were denoised, paired-end sequences were spliced, chimeric sequences were removed, and, finally, 463,856 valid data were obtained. The end of the sparse curve gradually flattened as the number of sequences per sample increased, indicating that the sample sequence is sufficient for data analysis (Figure 2). After the clustering threshold reached 97%, a total of 1563 OTUs were obtained, which were divided into 45 phyla, 110 classes, 296 orders, 571 families, 1204 genera and 1407 species.

### 3.2. Microbial Composition Analysis

Analysis of microbial community composition, revealed that that the main components of the pus samples from the same body parts were similar. At the genus level, the dominant bacterial genera in visceral pus sample P1 were *Trueperella* and *Bacteroides*, with relative abundances of 32.82% and 48.23%, respectively. The dominant bacterial genera in samples P2 and P3 were *Trueperella*, with relative abundances of 95.38% and 84.44%, respectively. The relative abundances of *Fusobacterium* in the body pus samples S2, S3, and S4 were 97.94%, 98.65%, and 99.52%, respectively (see Table 2). These results suggest that the dominant bacteria in the lung pus were mainly *Trueperella* and *Bacteroides*, while the dominant bacteria in the body surface pus were mainly *Fusobacterium* (Figure 3A).

The relative abundance of dominant pathogens at the species level was found to be almost identical to the abundance of their corresponding genera in different samples. *Trueperella pyogenes* and *Bacteroides fragilis* were the dominant pathogens in lung pus, while *Fusobacterium necrophorum* was the main pathogen in surface pus (Figure 3B). In the visceral pus sample P1, the dominant pathogens were *Trueperella pyogenes* and *Bacteroides fragilis,* with relative abundances of 32.82% and 48.21%, respectively. *Trueperella pyogenes* was the dominant pathogen in samples P2 and P3, with relative abundances of 95.38% and 84.44%, respectively. The relative abundance of *Fusobacterium necrophorum* in S2, S3, and S4 of the body surface pus samples was 97.94%, 98.65%, and 99.52%, respectively (Table 3). Table 4 shows the relative abundance of three dominant pathogens in six pus samples. The relative abundance of *Fusobacterium necrophorum* in visceral pus was 0, while the abundance of *Trueperella pyogenes* in the body surface pus was 0.

### 3.3. Correlation Analysis of Microbial Community Composition

Figure 4 shows the correlation between pathogens at the species level. Among the detected pathogens, *Fusobacterium necrophorum* and *Trueperella pyogenes* were significantly negatively correlated (*p* < 0.01). *Trueperella pyogenes* was not correlated with other bacteria. *Fusobacterium necrophorum* was negatively correlated with two unknown bacteria, but the correlation was not significant, indicating primarily single pathogenic bacteria causing forest musk deer abscess disease. Table 5 shows the correlation coefficients between microorganisms at the genus and species levels, respectively. There was no direct significant correlation between the dominant pathogenic bacteria and other bacteria in the samples, indicating that the dominant pathogenic bacteria played a major role in the process of lesion.

Microorganisms with the top 60 relative abundance in body surface and visceral pus were selected to screen data with a correlation greater than 0.1 and *p* value less than 0.05 to construct a correlation network for correlation analysis (Figure 5). Both *Trueperella* (Figure 5A) and *Fusobacterium* (Figure 5B) have other positively correlated pathogenic bacteria. Due to the low relative abundance, this article does not list them all. 

## 4. Discussion

The accurate diagnosis and identification of the main pathogenic bacteria that causes forest musk deer abscess can provide accurate guidance for the use of clinical drugs, reduce the use of unnecessary antibiotics, achieve rapid treatment of diseases and reduce the resistance of microorganisms to antibiotics in the body [12]. On the basis of previous studies, this study summarized the possible defects in the methods of existing studies, and used 16S rRNA high-throughput sequencing technology to detect the species and abundance of pathogenic bacteria in the pus of forest musk deer abscess at the genus level. The sequencing results of six pus samples showed that the dominant pathogenic bacteria in the pus of the same individual were relatively uniform, and the relative abundance in the flora was extremely high. In such cases, abscesses can be attributed to lesions in the body caused by specific bacteria. There were three dominant pathogenic bacteria detected in this study, namely *Trueperella pyogenes*, *Fusobacterium necrophorum* and *Bacteroides fragilis*. *Trueperella pyogenes* has been confirmed by researchers to be one of the pathogenic bacteria causing forest musk deer abscess [18]. Because *Fusobacterium necrophorum* and *Bacteroides fragilis* are obligate anaerobic bacteria [19,20], which cannot be cultured and proliferate by solid medium under aerobic conditions, these two bacteria could not be detected by the research methods reported in the existing studies. So far, there is no report of these two bacteria as the main pathogenic bacteria in forest musk deer abscesses. Our results demonstrate that diagnosis of pathogenic bacteria in abscess disease by high-throughput sequencing technology can not only overcome limitations of bacterial culture and improve the sensitivity of detecting known pathogenic bacteria, but also detect other potential pathogenic bacteria.

*Fusobacterium necrophorum* is one of the common anaerobic bacteria that can cause abscesses and respiratory infections in animals [21]. Common diseases caused in commercial animals are liver abscess [22], endometritis [23] and foot rot [24] in cattle and sheep, while *Fusobacterium necrophorum* has also been reported in wild animals such as North American bighorn sheep [25], white-tailed deer [26,27], tundra caribou and alpacas [28]. This study is the first to discover forest musk deer abscess disease caused by *Fusobacterium necrophorum* as the dominant pathogen. *Fusobacterium necrophorum* infection has a certain seasonality, often occurring in rainy, humid and hot seasons [29]. Therefore, it can be speculated that the main pathogenic bacteria in the three body-surface pus collected in summer is *Fusobacterium necrophorum*, but the reason why there is no *Fusobacterium necroptosis* in the pus samples collected in winter may be due to the influence of seasons. As an obligate anaerobic bacterium, *Fusobacterium necrophorum* overcomes the high oxygen concentration environment of the body organs and the phagocytosis mechanism of the body’s immune system through the cytotoxins secreted by itself and other facultative bacteria in the process of infecting the body [30]. Among them, leukotoxin is considered to be the main pathogenic factor of *Fusobacterium necrophorum* infection [31], and it is specific to neutrophils of ruminants and humans [32]. During the infection of the body, the leukotoxin and endotoxin secreted by *Fusobacterium necroptosis* act synergistically to produce toxicity to phagocytes and protect them from being swallowed [30]. Self-secreted leukotoxins cooperate with a series of virulence factors, such as platelet aggregation factors, to produce fibrin structures that can protect bacteria from phagocytosis by immune cells [33]. The hemolysin secreted by itself can destroy red blood cells and affect the oxygen transport capacity [34]. Meanwhile, the synergistic action of *Fusobacterium necrophorum* and other facultative bacteria facilitate secretion of endotoxins and platelet aggregation factors that induce intravascular coagulation [35], establishing the anaerobic microenvironment suitable for the growth of clostridium necrosis. As a result, *Fusobacterium necrophorum* infection on the surface of forest musk deer may be attributed to the individual injury of forest musk deer in a dark and humid enclosure. Factors such as the hot temperature in summer, the dark and humid environment in the enclosure, and residual feces and urine provide opportunities for *Fusobacterium necrophorum* to infect the body. Due to the small number of samples in this study and of only pus samples from the same body part, collected in the same season, this inference cannot be confirmed by the experimental data. The collection quantity can be further increased to compare the bacterial community structure of the pus samples from different parts of the body in the same season and the bacterial community structure of the pus samples from the same part of the body in different seasons to determine whether the main pathogenic bacteria types of the affected individuals are affected by the season, environment and lesion site.

*Bacteroides fragilis*, as an opportunistic pathogen, can also secrete a variety of pathogenic factors. When the host mucosa is damaged, it can invade the submucosa and cause infection. It can also cause suppurative infection accompanied by abscess and acute and chronic diarrhea through increased blood flow to other organs of the body, such as intestinal tract, abdominal cavity, liver, lung and brain tissue [36].

According to a published review, *Trueperella pyogenes* is the most common facultative bacterium that can coinfect with *Fusobacterium necrophorum* [19]. However, the results of this study show a highly significant negative correlation between the two. The reason for the lack of detection of *Fusobacterium necrophorum* infection in the visceral pus samples may be due to the seasonal sampling effect or the absence of infection by the pathogen in the host organism. The reason for the dominance of *Fusobacterium necrophorum* in the body surface pus samples may be due to the longer duration of the disease, as the anaerobic environment of the affected area is no longer suitable for the survival of other bacteria during the initial stage of the lesion. This leads to the survival and proliferation of *Fusobacterium necrophorum* in a suitable environment, causing infection and disease in the host organism.

In summary, this study used high-throughput sequencing technology to detect the types of pathogenic bacteria in the pus of forest musk deer abscess disease at the genetic level. In addition to confirming the presence of recognized pathogens, previously unreported pathogenic bacteria were also discovered, providing a new approach to the diagnosis of abscess disease pathogens. Due to limitations in sample size and sequencing fragment length, the number of detected pathogenic bacteria in the results was relatively small. Moreover, the use of 16SrRNA sequencing technology can only describe bacterial populations and cannot detect the presence of pathogens such as viruses. 

Given this, in future research, the sequencing fragment length can be increased to analyze and display more pathogenic bacteria at the species level. Secondly, the number of samples for detection can be expanded to improve the types of pathogenic bacteria that may exist, providing a basis for clinical diagnosis and drug use. Finally, with the development of high-throughput sequencing technology and the decrease in sequencing costs, metagenomic sequencing technology can be used to improve the pathogen types of abscess disease and analyze the possible pathogenic mechanisms of different pathogens from the perspective of microbial function, providing a reference for further drug and vaccine development.

## 5. Conclusions

In terms of pathogen detection, high-throughput sequencing technology can overcome the limitations of bacterial culture for detection of pathogens. Two pathogens, *Fusobacterium necrophorum* and *Bacteroides fragilis*, were detected for the first time in forest musk deer abscesses by using this technology.

## Figures and Tables

**Figure 1 animals-13-03142-f001:**
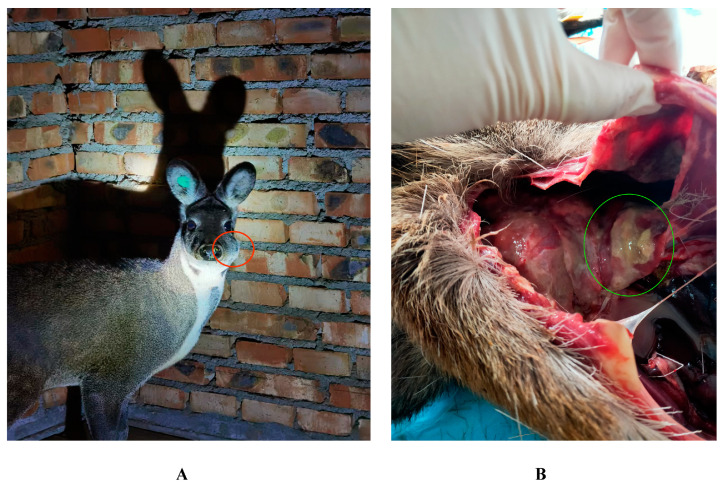
Characteristic lesion locations of forest musk deer abscess disease. (**A**) Surface lesions; (**B**) Visceral lesions. The red circle in Figure A represents the site of surface lesions, and the green circle in Figure B represents the site of visceral lesions.

**Figure 2 animals-13-03142-f002:**
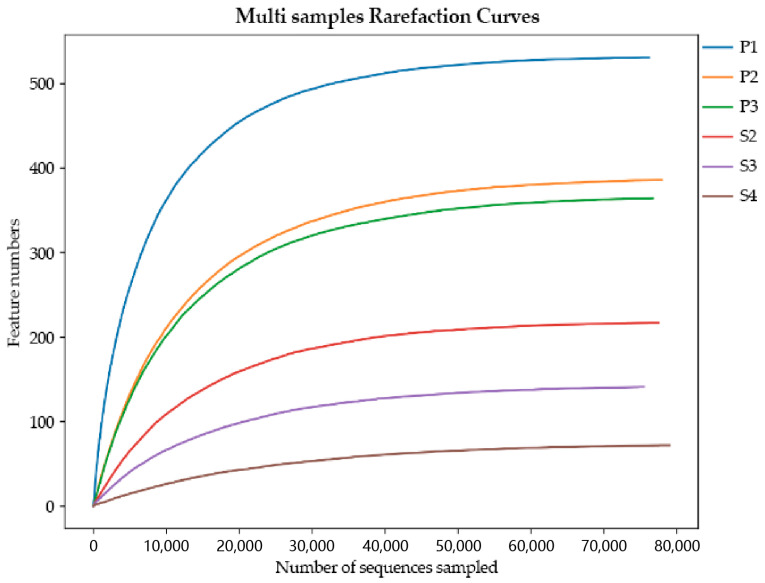
Rarefaction curves. The *X*-axis is the number of randomly selected sequencing entries, and the *Y*-axis is the number of features obtained based on the number of sequencing entries. Each curve represents a sample and is marked with a different color.

**Figure 3 animals-13-03142-f003:**
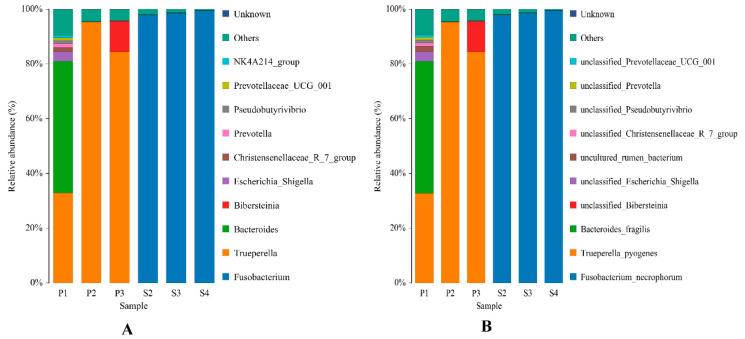
Composition and relative abundance of bacterial flora in different pus samples at the genus (**A**) and species (**B**) levels.

**Figure 4 animals-13-03142-f004:**
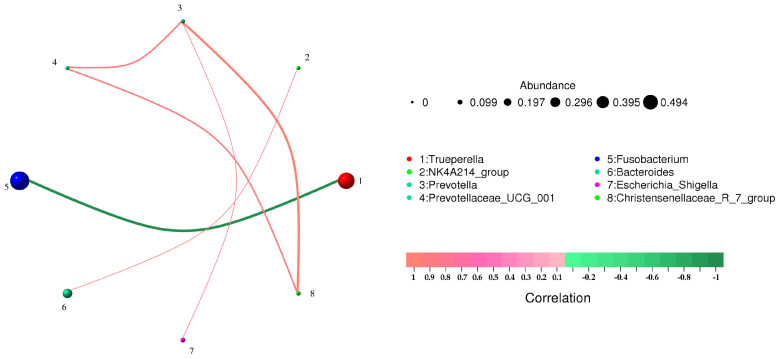
Correlation network analysis of bacterial structure in different pus samples at species level. Circles represent species, and the size of circles represents the average abundance of species; the lines represent the correlation between the two species, and the thickness of the lines represents the strength of the correlation; line colors: red for positive correlation, green for negative correlation.

**Figure 5 animals-13-03142-f005:**
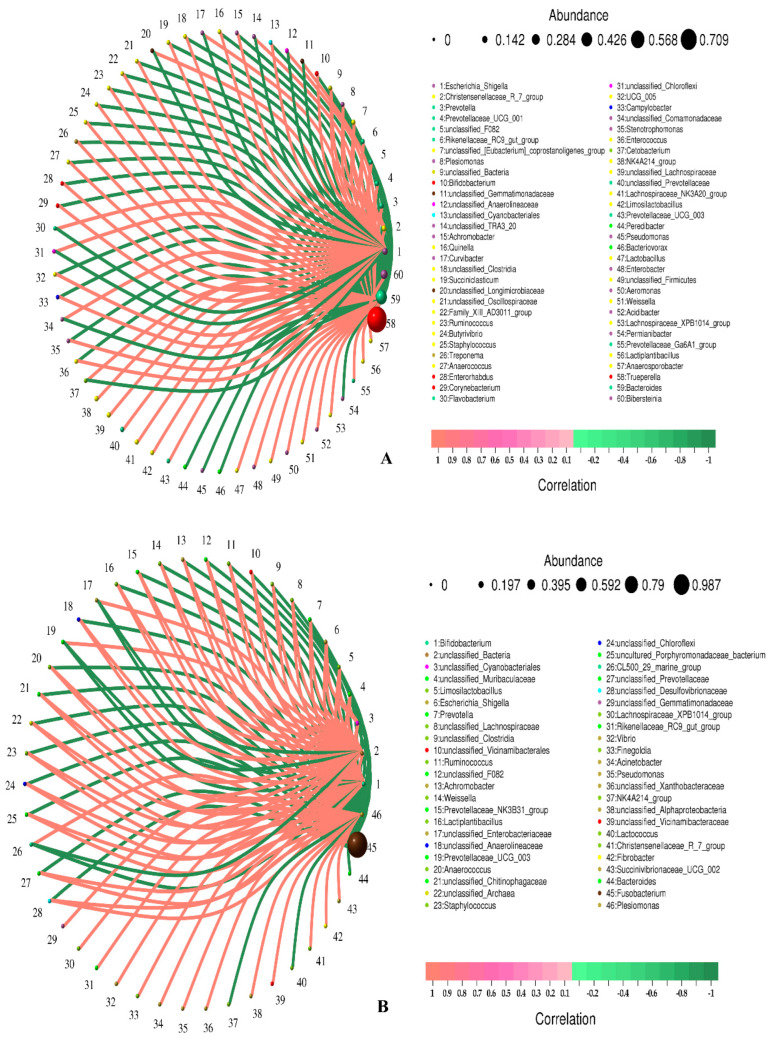
Correlation network analysis of bacterial structure in visceral (**A**) and superficial (**B**) pus samples at the species level. Circles represent species, and the size of circles represents the average abundance of species; the lines represent the correlation between the two species, and the thickness of the lines represents the strength of the correlation; line colors: red for positive correlation, green for negative correlation.

**Table 1 animals-13-03142-t001:** Pus sample information.

Lesion	Sex	Strain No
Lung	Male	P1
Lung	Male	P2
Lung	Male	P3
Hoof	Male	S2
Hoof	Male	S3
Face	Female	S4

**Table 2 animals-13-03142-t002:** The main pathogenic bacteria and their relative abundance at the genus level.

Simple	Species and Abundance at the Generic Level
P1	*Trueperella* (32.82%) *Bacteroides* (48.23%)
P2	*Trueperella* (95.38%)
P3	*Trueperella* (84.44%)
S2	*Fusobacterium* (97.94%)
S3	*Fusobacterium* (98.65%)
S4	*Fusobacterium* (99.52%)

**Table 3 animals-13-03142-t003:** The main pathogenic bacteria and their relative abundance at the species level.

Simple	Species and Abundance at the Species Level
P1	*Trueperella pyogenes* (32.82%) *Bacteroides fragilis* (48.21%)
P2	*Trueperella pyogenes* (95.38%)
P3	*Trueperella pyogenes* (84.44%)
S2	*Fusobacterium necrophorum* (97.94%)
S3	*Fusobacterium necrophorum* (98.65%)
S4	*Fusobacterium necrophorum* (99.52%)

**Table 4 animals-13-03142-t004:** Relative abundance of 3 dominant pathogens in 6 pus samples.

Pathogenic Bacteria.	P1	P2	P3	S2	S3	S4
*Fusobacterium_necrophorum*	0	0	0	97.94%	98.65%	99.52%
*Trueperella_pyogenes*	32.82%	95.38%	84.44%	0.013%	0	0
*Bacteroides_fragilis*	48.21%	0	0	0	0	0

**Table 5 animals-13-03142-t005:** Correlation coefficient at genus level.

From	To	Corr	*p* Value
*Trueperella*	*Fusobacterium*	−0.985610761	0.000309086
*NK4A214_group*	*Bacteroides*	0.828571429	0.041562682
*Prevotella*	*Escherichia_Shigella*	0.828571429	0.041562682
*Prevotella*	*Christensenellaceae_R_7_group*	0.942857143	0.004804665
*Prevotellaceae_UCG_001*	*Christensenellaceae_R_7_group*	0.845154255	0.034109423
*Prevotellaceae_UCG_001*	*Prevotella*	0.845154255	0.034109423

From is the name of the first node; To is the name of the second node; Corr is the correlation between two nodes. The larger the value, the stronger the correlation; *p* value is a significant correlation between the two, *p* < 0.05 is a significant correlation, *p* < 0.01 is an extremely significant correlation.

## Data Availability

The datasets presented in this study can be found in online repositories.

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
