# Peer review of "Detecting Forest Musk Deer Abscess Disease Pathogens Using 16S rRNA High-Throughput Sequencing Technology"

_animals, 2023, doi:10.3390/ani13193142_

Round 1

Reviewer 1 Report

This preliminary study used 16S high throughput sequencing technology to determine the microbial content in 3 visceral and 3 surface abscess lesions from musk deer, an important type of Chinese livestock.  The authors do a great job introducing the disease and the need for the work.

Trueperella pyogens, a facultative anaerobic pathogen previously determined to be the causative agent in in musk deer abscess disease by culture, was detected in the three visceral abscesses and was the predominant pathogen detected in 2 of these. Bacteroides fragilis, an anaerobe which is considered an opportunistic pathogen, was also detected to high levels in the third.  Fusobacterium necrophorum, an obligate anaerobe was the predominant pathogen in the 3 studied surface abscess lesions.  Although F. necrophorum has not previously been identified in musk deer abscesses, it is a common animal pathogen known to cause abscess disease.  The authors recognize and discuss this as a benefit of this sequencing method over routine aerobic culture.  Potential limitations of the study- small sample size and the short falls of the method for detecting other pathogens such as viruses were mentioned. 

Suggested Improvements

Although the findings show the potential of the method, this reviewer feels there should be more emphasis on the preliminary nature of the work.  The use of this method for pathogen detection is relatively novel.  There were over 130 diseased samples characterized in the cited reference describing the use of this technology for screening mastitic milk and these were compared by both culture and sequencing and there were samples of uninfected milk characterized as well.  If possible, a larger sample size with comparison using aerobic and anaerobic culture would help to validate the use of 16S high throughput sequencing for routine diagnosis of the cause of abscesses in musk deer.  Without growth and susceptibility testing, it was also not clear how this method would facilitate determination of proper antibiotic treatment.  There would be some value in discussing this.

The striking uniform nature of these samples would be more believable if repeated with a second 16S hypervariable region in addition to V3,V4, since there is evidence that this can significantly vary the observed distribution of observed bacteria- see R. Lopex-Aladid et al Scientific Reports (13) article 3974.

Specific corrections

Since samples P4, P5 and S1 were likely to have been contaminated, they were therefore not collected aseptically as described in the methods sections and should be left out since they simply confuse things as is.

Since Figure 3A and B appear identical, it seemed a waste and confusing to have both, especially since most of the listed bacteria in B are not identified to the species level.  It would make sense to combine A and B and note which single pathogen species were detected in the figure legend.

 Given that there was no obvious difference between 3A and B, the significance of the differences between results in Figure 4 and Figure 5 seem questionable and confusing.  Perhaps it would make sense to leave one or the other out.  Also, all software methods should be referenced.

Although the article is well written and well referenced and the English is understandable, there is room for improvement.  In particular, sections 2.3 and 2.4 in the methods section should state what was done in the past tense rather than as it is currently, for example PCR products were (not “are”) used to form…

Author Response

Response to Reviewer 1’s Comments

1. Summary

On behalf of my co-authors, we thank you very much for giving us an opportunity to revise our manuscript. We are very grateful to you for the positive and constructive comments and suggestions on our entitled manuscript entitled “Detecting Forest musk deer (Moschus berezovskii) abscess disease Pathogenic bacteria using 16S rRNA high-throughput sequencing technology”. We have studied your comments carefully and have made revisions marked in manuscript. We tried our best to improve the manuscript and made some changes in the manuscript. These changes will not affect the content and framework of the paper. We hope that our revision could address all of the questions. We are looking forward to hearing from you soon.

Best wishes,

                    Yours sincerely

2. Questions for General Evaluation

Reviewer’s Evaluation

Response and Revisions

Does the introduction provide sufficient background and include all relevant references?

Yes

Are all the cited references relevant to the research?

Yes

Is the research design appropriate?

Can be improved

Are the methods adequately described?

Can be improved

Are the results clearly presented?

Can be improved

Are the conclusions supported by the results?

Yes/Can be improved/Must be improved/Not applicable

3. Point-by-point response to Comments and Suggestions for Authors

Comments 1: Although the findings show the potential of the method, this reviewer feels there should be more emphasis on the preliminary nature of the work.  The use of this method for pathogen detection is relatively novel.  There were over 130 diseased samples characterized in the cited reference describing the use of this technology for screening mastitic milk and these were compared by both culture and sequencing and there were samples of uninfected milk characterized as well.  If possible, a larger sample size with comparison using aerobic and anaerobic culture would help to validate the use of 16S high throughput sequencing for routine diagnosis of the cause of abscesses in musk deer. Without growth and susceptibility testing, it was also not clear how this method would facilitate determination of proper antibiotic treatment.  There would be some value in discussing this.

Response 1: Thank you greatly for your valuable suggestion. The primary objective of this study is to demonstrate that the outcomes achieved through 16S rRNA sequencing technology, during the diagnosis of pathogenic bacteria causing forest musk deer abscess disease, offer a broader perspective compared to those attained through traditional culture techniques. This research serves as a foundation research in antibiotic screening and pharmaceutical development, as highlighted in the introduction (Lines 74 – 84).

In this research, the application of 16S rRNA sequencing technology in forest musk deer abscess disease not only corroborated the presence of Trueperella pyogens, as previously noted by researchers, but also unveiled two previously unidentified pathogenic bacteria. This underscores the technological advantage of this method in identifying pathogenic bacteria. As you correctly pointed out, the growth and susceptibility testing form the basis for our forthcoming research phase, which builds upon these discoveries.

Simultaneously, it's important to note that the sample preservation requisites for growth and susceptibility testing vary from those essential for high-throughput sequencing technology. Consequently, we must adapt the sample preservation conditions based on the outcomes derived from high-throughput sequencing during the sample collection process. For the objectives of this study, Whether we can consider growth and susceptibility tests is outside the scope of this study and look forward to your valuable suggestions.

Comments 2: The striking uniform nature of these samples would be more believable if repeated with a second 16S hypervariable region in addition to V3, V4, since there is evidence that this can significantly vary the observed distribution of observed bacteria- see R. Lopex-Aladid et al Scientific Reports (13) article 3974.

Response 2: Thank you very much for your suggestion. In this study, we aim to prove that 16S rRNA sequencing technology can overcome the limitations of traditional culture medium. We chose the V3-V4 region for detection mainly because the technology in this segment is relatively mature and inexpensive. After confirming the practicality of high-throughput sequencing technology, our further research will consider detecting the full length of 16s or using metagenomic technology for further study. This study can achieve our experimental purpose by detecting the V3 -V4 region.

Comments 3: Since samples P4, P5 and S1 were likely to have been contaminated, they were therefore not collected aseptically as described in the methods sections and should be left out since they simply confuse things as is.

Response 3: Your suggestion is greatly appreciated. We have rewritten it(Line 96 - 102).

Comments 4: Since Figure 3A and 3B appear identical, it seemed a waste and confusing to have both, especially since most of the listed bacteria in B are not identified to the species level.  It would make sense to combine A and B and note which single pathogen species were detected in the figure legend.

Response 4: Thank you very much for your suggestion. In Figure 3, A and B represent the relative abundance of each sample genus and species respectively. The purpose of this is to make a clear comparison. From Figures 3A and 3B, in each sample, the relative abundance of the main pathogenic bacteria at the genus level is consistent with the relative abundance at the species level (Table 2 and Table 3 also show the genus level of each group of pathogenic bacteria and relative abundance at the species level). The conclusion drawn from this result is that the main pathogenic bacteria in each group of samples is single, and there are no other pathogenic bacteria in the same genus.  

Comments 5: Given that there was no obvious difference between 3A and 3B, the significance of the differences between results in Figure 4 and Figure 5 seem questionable and confusing.  Perhaps it would make sense to leave one or the other out.  Also, all software methods should be referenced.

Response 5: Thank you very much for your suggestion. We have deleted Figure 4 in the manuscript and revised Figure 5 to Figure 4. All software methods have been referenced in the manuscript.I apologize for this low-level mistake, and thank you for your correction.

4. Response to Comments on the Quality of English Language

Point 1: Although the article is well written and well referenced and the English is understandable, there is room for improvement.  In particular, sections 2.3 and 2.4 in the methods section should state what was done in the past tense rather than as it is currently, for example PCR products were (not “are”) used to form…

Response 1: Thank you very much for your suggestion. We have made corresponding changes.We apologize for this low-level mistake, and thank you for your correction.

Reviewer 2 Report

This paper describes and new and valuable diagnostic tool and identifies new pathogens for abscesses of forest musk deer. This deer species is critically endangered and abscess disease is an important cause of morbidity and mortality.

The paper is scientifically sound. My main suggestions regard some confusing terminology and general improvements.

The pathogen “Fusobacterium necrophorum” is referred to by multiple different names including “Fusobacterium necrophorum”, “Fusobacterium necroptosis”, and “Fusobacterium necrosis”. The authors need to be consistent and use the proper term which I believe is still Fusobacterium necrophorum, unless it has recently changed (if so, the authors may want to note this).

The pictures in figure 1 would benefit from being enlarged.

Why are the Case numbers listed in Table 1? I believe they could be deleted.

Some of the descriptions in the Methods and Results sections (especially 2.316, 2.4 and 3.3) appear to be taken directly from the EZNA manufacturer’s instructions or they may have been developed by Artificial Intelligence. In any case, they should describe what the authors did, and found, not provide instructions on how to do it.  

There was great detail and discussion on correlation of bacteria at the genus and species level. I did not find this particularly useful. Some of this discussion could be reduced at the authors’ discretion.  

In general, the quality of English was acceptable. There are times when the paper reads like it was written using Artificial Intelligence. I have no issue with using AI to help write papers, but authors must ensure the product is accurate and sensible. My specific recommendations:

As mentioned above, be consistent with the name of Fusobacterium necrophorum.

Line 23: clarify that Fusobacterium necrophorum and Bacteroides fragilis (not “they”) along with Truperella pyogenes are dominant pathogens.

Line 29: I think you mean “genetic” not “gene”

Line 60: I believe you are specifically referring to antibiotics for internal abscesses since skin ones can be surgically removed.

Line 91: clarify if these were farmed (as I believe they were) or free-range deer.

Line 181: Table 2 should start on a new page instead of having the heading on one page and the body on another. Also, “Sample” is misspelled as “Simple”.

Line 213. Replace the comma between “significant” and “It” with a period.

Lines 218-241: Since Table 5 is discussed in the text before Figure 6, Table 5 should appear before Figure 6.

Line 244: replace “abscess” with “abscesses.”

Line 249: replace “gene” with “genetic”.

Line 251: Relace “relatively” with “mostly”. And replace “in” with “of”.

Line 259: Delete “research”.

Lines 270-271 and 274-275: Bacteria names should be italicized.  

Line 282: Hyphen should be deleted.

Line 290: Replace “clostridium necrosis” with a particular Clostridium species.

Author Response

Response to Reviewer 2’s Comments

1. Summary

On behalf of my co-authors, we thank you very much for giving us an opportunity to revise our manuscript. We are very grateful to you for the positive and constructive comments and suggestions on our entitled manuscript entitled “Detecting Forest musk deer (Moschus berezovskii) abscess disease Pathogenic bacteria using 16S rRNA high-throughput sequencing technology”. We have studied your comments carefully and have made revisions marked in manuscript. We tried our best to improve the manuscript and made some changes in the manuscript. These changes will not affect the content and framework of the paper. We hope that our revision could address all of the questions. We are looking forward to hearing from you soon.

Best wishes,

                    Yours sincerely

2. Questions for General Evaluation

Reviewer’s Evaluation

Response and Revisions

Does the introduction provide sufficient background and include all relevant references?

Yes

Are all the cited references relevant to the research?

Yes

Is the research design appropriate?

Yes

Are the methods adequately described?

Can be improved

Are the results clearly presented?

Can be improved

Are the conclusions supported by the results?

Yes

3. Point-by-point response to Comments and Suggestions for Authors

Comments 1: The pathogen “Fusobacterium necrophorum” is referred to by multiple different names including “Fusobacterium necrophorum”, “Fusobacterium necroptosis”, and “Fusobacterium necrosis”. The authors need to be consistent and use the proper term which I believe is still Fusobacterium necrophorum, unless it has recently changed (if so, the authors may want to note this).

Response 1: Thank you very much for your suggestion. We modified the name of Fusobacterium necrophorum in the whole paper to be consistent with the whole paper, and marked the modified content in red in the paper.I apologize for this low-level mistake, and thank you for your correction.

Comments 2: The pictures in figure 1 would benefit from being enlarged.

Response 2: Thank you very much for your suggestion. We have made corresponding changes.

Comments 3: Why are the Case numbers listed in Table 1? I believe they could be deleted.

Response 3: Your suggestion is greatly appreciated. And we have deleted it.

Comments 4: Some of the descriptions in the Methods and Results sections (especially 2.316, 2.4 and 3.3) appear to be taken directly from the E.Z.N.A manufacturer’s instructions or they may have been developed by Artificial Intelligence. In any case, they should describe what the authors did, and found, not provide instructions on how to do it.

Response 4: Thank you very much for your suggestion. We have made corresponding changes in the manuscript.

Comments 5: There was great detail and discussion on correlation of bacteria at the genus and species level. I did not find this particularly useful. Some of this discussion could be reduced at the authors’ discretion.  

Response 5: Thank you very much for your suggestion. The purpose of our detailed discussion of the significant negative correlation between the two pathogenic bacteria in different samples at the genus and species level is to show that the main pathogenic bacteria of different infected individuals obtained by 16s rRNA sequencing are different from the conclusions of previous studies, and further reflect the comprehensiveness of the detection of pathogens of forest musk deer abscess by 16s rRNA sequencing technology.This is why we discuss this section in detail and look forward to your valuable suggestions.

4. Response to Comments on the Quality of English Language

Point 1: As mentioned above, be consistent with the name of Fusobacterium necrophorum.

Response 1: Thank you very much for your suggestion. We modified the name of Fusobacterium necrophorum in the whole paper to be consistent with the whole paper, and marked the modified content in red in the paper. We apologize for this low-level mistake, and thank you for your correction.

Point 2: Line 23: clarify that Fusobacterium necrophorum and Bacteroides fragilis (not “they”) along with Truperella pyogenes are dominant pathogens.

Response 2: Thank you very much for your suggestion. We have modified “they” to “Fusobacterium necrophorum and Bacteroides fragilis”, and marked the modified content in red in the paper (Line 24).

Point 3: Line 29: I think you mean“genetic” not“gene”

Response 3: Thank you very much for your suggestion.We have modified “gene” to “genetic”, and marked the modified content in red in the paper (Line 30).

Point 4: Line 60: I believe you are specifically referring to antibiotics for internal abscesses since skin ones can be surgically removed.

Response 4: We are very sorry that we did not express ourselves clearly in the article. In clinical practice, the skin abscess of forest musk deer needs antibiotics to inhibit further infection of pathogenic bacteria after surgical resection, while the internal abscess can only be treated with antibiotics. Therefore, the use of broad-spectrum antibiotics is for individuals with abscess with all symptoms, which we have modified in the paper and highlighted in red (Line 60 - 61).

Point 5: Line 91: clarify if these were farmed (as I believe they were) or free-range deer.

Response 5: Thank you very much for your suggestion. We added "farmed" in the paper and highlighted it in red (Line 93). 

Point 6: Line 181: Table 2 should start on a new page instead of having the heading on one page and the body on another. Also, “Sample” is misspelled as “Simple”.

Response 6: Thank you very much for your suggestion. We have adjusted the contents of Table 2 on a piece of paper and modified the word "Simple" to "Sample" (Line 183 - 184).

Point 7: Line 213. Replace the comma between “significant” and “It” with a period.

Response 7: Thank you very much for your suggestion. We have replaced the comma between “significant” and “It” with a period and highlighted it in red (Line 215).

Point 8: Lines 218-241: Since Table 5 is discussed in the text before Figure 6, Table 5 should appear before Figure 6

Response 8: Thank you very much for your suggestion. We have moved Table 5 ahead of Figure 6 (Line 227 – 231).

Point 9: Line 244: replace “abscess” with “abscesses.”

Response 9: Thank you very much for your suggestion. We have replaced “abscess” with “abscesses” in the paper and highlighted it in red (Line 249).

Point 10: Line 249: replace “gene” with “genetic”.

Response 10: Thank you very much for your suggestion. We have replaced “gene” with “genetic” in the paper and highlighted it in red (Line 254).

Point 11: Line 251: Relace “relatively” with “mostly”. And replace “in” with “of”.

Response 11: Thank you very much for your suggestion. We have replaced “relatively” with “mostly”. And replace “in” with “of”. And highlighted them in red (Line 256).

 Point 12: Line 259: Delete “research”.

Response 12: Thank you very much for your suggestion. We have deleted “research” (Line 264).

 Point 13: Lines 270-271 and 274-275: Bacteria names should be italicized. 

Response 13: Thank you very much for your suggestion. We have made corresponding changes.

 Point 14: Line 282: Hyphen should be deleted.

Response 14: Your suggestion is greatly appreciated. And we deleted it.

 Point 15: Line 290: Replace “clostridium necrosis” with a particular Clostridium species.

Response 15: Thank you very much for your suggestion. We have replaced “clostridium necrosis” with ” Fusobacterium necrophorum” and highlighted it in red (Line 294).

Round 2

Reviewer 1 Report

I am happy with the adjustments/improvements made by the authors.  

There were several instances where I felt the text needed some clarification/adjustment.  I have attached suggestions for this clarificationh.

Author Response

Response to Reviewer 1 Comments

1. Summary

On behalf of my co-authors, we thank you very much for your language suggestions, we are sorry for our simple mistakes. We have made corresponding modifications in the paper according to your suggestions and marked them with red ink. We hope that our revision could address all of the questions. We are looking forward to hearing from you soon.

Best wishes,

                    Yours sincerely

2. Questions for General Evaluation

Reviewer’s Evaluation

Response and Revisions

Does the introduction provide sufficient background and include all relevant references?

Yes

Are all the cited references relevant to the research?

Yes

Is the research design appropriate?

Yes

Are the methods adequately described?

Yes

Are the results clearly presented?

Yes

Are the conclusions supported by the results?

Yes